# New Manufacturing Process of Composites Reinforced with ZnO Nanoparticles Recycled from Alkaline Batteries

**DOI:** 10.3390/polym12071619

**Published:** 2020-07-21

**Authors:** Isaac Lorero, Mónica Campo, Gilberto Del Rosario, Félix Antonio López, Silvia González Prolongo

**Affiliations:** 1Materials Science and Engineering Area, Rey Juan Carlos University, C/Tulipán s/n, 28933 Móstoles, Madrid, Spain; monica.campo@urjc.es; 2Laboratory for Electron Microscopy, Centre for Technical Support (CAT), Rey Juan Carlos University, C/Tulipán s/n, 28933 Mostoles, Madrid, Spain; gilberto.delrosario@urjc.es; 3National Centre for Metallurgical Research (CENIM), Spanish National Research Council (CSIC), Avda, Gregorio del Amo, 8. 28040 Madrid, Spain; flopez@cenim.csic.es

**Keywords:** nanocomposites, epoxy, zinc oxide, sedimentation, recycled alkaline batteries

## Abstract

A new manufacturing method of thermosetting resins reinforced with dense particles is developed in the present work. A rotary mold is used, avoiding the natural sedimentation of particles through applying centrifuge forces. A deep study of the sedimentation phenomenon is carried out in order to evaluate the main experimental parameters which influence the manufacturing of composite. The used reinforcement is zinc oxide (ZnO) obtained by a new recycling method from spent alkaline batteries. In order to compare the benefits, commercial ZnO nanoparticles are also analyzed. Recycled ZnO particles enhance the interaction of the epoxy matrix due to their inner moisture, allowing the manufacture of composites with relatively high ceramic content. Moreover, an increment in the glass transition temperature of the epoxy matrix and in the mechanical properties, such as its stiffness and hardness, is achieved.

## 1. Introduction

The manufacture of new materials with hydrophobic polymer surfaces has attracted increasing attention in recent years. Capabilities like self-cleaning, anti-icing, corrosion inhibition, or hydrothermal ageing resistance can be considered among the properties derived from water and moisture repellent surfaces [1,2,3,4,5,6]. This behavior depends on several experimental parameters, such as the geometry of the surface, measured by superficial roughness, and the chemistry of materials. Due to its intrinsic characteristics, many thermosets commonly used in the industry, as the epoxy resins for manufacturing composites and coatings, present a hydrophilic nature. One way to enhance the hydrophobicity of these matrix surfaces is the introduction of dispersed ceramic nanoparticles, which increase its roughness. The application of different post-treatments can reduce the surface energy and contribute to create hierarchical micro and nanostructures [7,8,9,10,11]. In addition to providing hydrophobicity, the ceramic nanoreinforcements contribute with other improvements: increase in mechanical properties when there is an effective adhesion with the matrix, increase of thermal stability, enhancement of hardness, and wear resistance among others [5,12,13,14,15,16]. Between the ceramic nanoreinforcements based on the metallic oxides, ZnO nanoparticles are widely studied because of their varied and exceptional properties for different technological applications, which make this material an appropriate selection to produce multifunctional materials [15,17,18,19,20,21]. One interesting attribute for its use as a reinforcement for resins is its high-quality dispersion and low tendency to form large aggregates within the matrix compared to other oxide nanoparticles [4,17]. Also, previous researches found that ZnO particles may catalyze the epoxy curing reaction, increasing the crosslinking density of the epoxy network [22,23,24]. Zinc oxide can introduce other advantages, such as the enhancement of antibacterial activity, piezoelectricity, corrosion and UV degradation resistance, increased thermal conductivity, or the capacitance for new structural supercapacitors in comparison with neat resins [17,25,26,27,28,29,30,31,32,33]. 

The use of alternative recycling synthesis to recover materials as ceramic oxides opens an interesting opportunity to create composites by more sustainable ways [34]. “Urban mining” is the recovery of metals and ceramics through mechanical and chemical treatments, providing ecological and effective reinforcements. For example, hydrometallurgical routes could generate particles with inherent moisture, being able to enhance the crosslinking density and the interaction with the matrix and thus positively influence thermomechanical composites properties [22,23]. 

On the other hand, the inherent moisture can also enhance the agglomeration phenomenon of particles, hindering the dispersion. In that way, ZnO particles, recovered from spent alkaline batteries, have been used in the present research as fillers of composites in order to compare its efficiency regard to commercial nanoparticles [35,36]. It is worthy to note that this work seeks new applications for recycled ZnO from batteries, whose mass storage as waste contributes to environmental and health problems. According to battery sales worldwide, the most used battery type is alkaline batteries (75%) due to their having a higher performance than other batteries. Each year, thousands of tons of alkaline batteries must be recycled, thus new use of the obtained recycled materials must be developed. 

Looking at the main factors to consider during the manufacturing of these composite materials, the high differences in density between ceramic fillers and polymer resins reveals one of the most critical points. This gap, attached to their high specific areas of fillers, which induces severe problems of aggregation, the relative low viscosity of epoxy monomers and large gel time during the curing process of thermosetting resins could lead to pauperize the reinforcement dispersions. This could cause the particles sedimentation to the bottom of composite, avoiding the potential improvements in material properties due to the heterogeneity of samples. Higher particle sizes also reveal as a decisive factor in these sedimentation phenomenon [37]. One way to favor the maintenance of ceramic filler dispersions within the matrix for the curing process is the functionalization of the particles by chemical treatments before mixing with the resin. Repulsive forces between functionalized particles and positive interactions with the polymer matrix prevent their agglomeration and sedimentation. However, these treatments imply some limitations for future industrial applications due to the use of unhealthy reagents and the increase of manufacturing times. Our current proposal consists of the neutralization of downfall movements of fillers into thermosetting resin through the removal of the gravity effect by applying mechanical movements. This solution provides easier and faster manufacturing routes to obtain well dispersed and effective composites. 

In the present research, a new manufacturing technique is optimized, creating one method able to leave out the pre-treatment of a particle’s functionalization through using a rotary mold during the curing cycle of composite. Thanks to the rotational movement, the sedimentation is avoided, and the reinforcement distribution achieved during the previous sonication dispersion is maintained, creating homogeneous composites with enhanced thermo-mechanical properties and hardness. 

As mentioned above, the main goal is to develop a new method to manufacture epoxy composites with recycled ZnO particles from spent alkaline batteries are used as fillers in comparison with commercial nanoparticles. The battery recycling and the reuse of obtained ZnO particles have important environmental and economic benefits. 

## 2. Experimental

### 2.1. Materials

Polymer matrix used in the present work was a two-component epoxy resin based on bisphenol A (Araldite LY 556) mixed with aromatic amine hardener (XB 3473) in a weight ratio of 100:23. Both components were purchase from Huntsman (Woodlands, TX, USA). In order to analyze the influence of ceramic filler characteristics on the epoxy properties, two different zinc oxides were selected: commercial nanoparticles with >98% purity supplied by Merck (Darmstadt, Germany) and alternative particles recovered from spent alkaline batteries [35]. The synthesis of this second type of zinc oxide consists on a three stages process: 

1. Leaching of batteries black mass: black mass was leached by adding 1.72 mol/L of (NH_4_)_2_CO_3_ and 0.5 mol/L of NH_3._ Black mass pulp concentration was settled at 100 g/L and the solution was agitated at 1800 rpm under ambient conditions for 1 h. The after-treatment suspension obtained was filtered under vacuum and leachate was kept for further processing.

2. Leachate evaporation under vacuum (34 mbar) and drying of the solid precursor gotten in an oven at 80 °C for 6 h.

3. Calcination of the post-evaporation solid precursor at 800 °C for 5 h to obtain the ZnO.

Finally, the manufactured composites were treated with successive chemical immersions to give them hydrophobicity. For that, sulfuric acid, ethanol, glacial acetic acid, and stearic acid are used as chemical solvents. All these reagents were supplied by Merck. It is worthy to note that all components were used as received and none of the particles were functionalized or treated before the composites manufacturing.

### 2.2. Composites Manufacturing

Neat epoxy resin, without filler, was manufactured and taken as reference of the research. On the other hand, epoxy composites were manufactured by adding different concentrations of commercial and recycled ZnO (2, 6, and 10 wt %). Finally, due to the obtained results, another sample with a 30 wt % of ZnO was used to analyze the influence of higher loads. 

To create the composites, the epoxy resin was heated up to 50 °C to reduce its viscosity and then the ZnO particles were added and mixed mechanically. To achieve a homogeneous distribution, the particles were dispersed afterwards into the epoxy matrix by sonication with a cycle of 50 Hz and 50% of amplitude during 1 h. This was done by using a 7 mm diameter probe sonicator. To control the temperature during sonication and avoid overheating capable of triggering epoxy homopolymerization, cold water baths were placed around the epoxy-ZnO mixtures. Afterwards, the dispersion was degassed 15 min at 80 °C. Then, the hardener was added in a stoichiometric ratio and finally the non-cured mixture was placed in a mold for curing at 140 °C for 8 h.

Composites were manufactured by two different molding methodologies. First, the materials were putted into an open mold, which remains static in the oven during the curing cycle. Due to the undesirable sedimentation of fillers, a rotary molding system were probed to avoid this phenomenon (Figure 1). In these cases, composites were introduced into a closed mold, which was screwed to a rotation shaft placed along the oven. The rotation shaft was actuated by an external electric engine to move the mold at 11 rpm in order to prevent the downward movement of ceramic particles. To avoid the air occlusion inside the rotary mold and the consequent porosity in cured composites, the mold must be slightly overfilled before closing and starting the curing cycle. 

### 2.3. Characterization

The morphology of the particles used as reinforcements were characterized by field emission scanning electron microscopy (FEG-SEM, Nova Nano SEM230 model, FEI, Hillsboro, OR, USA). 

ZnO particles were analyzed by Fourier transform infrared spectroscopy (FTIR), using a Varian 670 FTIR spectrometer (Varian Inc., Palo Alto, CA, USA) (spectral range 4000–400 cm^−1^, spectral resolution of 4 cm^−1^) in transmittance mode. This analysis was performed using KBr pellets with 1 wt % of ZnO. Epoxy-ZnO composites were also analyzed by Attenuated total reflectance-Fourier transform infrared spectroscopy (ATR-FTIR) (Varian Inc., Palo Alto, CA, USA) with Nicolet (Termo Fischer Scientific Inc., Waltham, MA, USA) ATR accesory (spectral range 3800–700 cm^−1^). The chemical composition of recycled ZnO was also analyzed by X-ray fluorescence (XRF) using a PANalytical Acios wavelength dispersive spectrometer (4 kW) (Malvern, Worcestershire, United Kingdom).

The dispersions of the zinc oxide in epoxy monomer were evaluated with a reflected light microscope (Transmission optical microscope, TOM, Leyca Microsystem M205 C, Weltzar Germany). These images were digitally analyzed to characterize the distribution and size of particles and aggregates. Later, the final dispersion on cured composites was analyzed by scanning electron microscopy (SEM, Hitachi S3400N, Tokyo, Japan). 

Thermomechanical properties of neat resin and composites were measured by dynamical thermo-mechanical analysis (DTMA) (New Castle, DE, USA) with a Q800 TA Instruments. Tests were made in single cantilever mode with an amplitude of 1% regard to samples thickness and a frequency of 1 Hz. Thermomechanical data were collected in a temperature range from 30 °C to 230 °C with a heating rate of 2 °C/min.

Composites Vickers micro hardness were measured by applying 980.7 mN loads with a Shimadzu HMV-2T indentation tester.

## 3. Results and Discussion

### 3.1. Characterization of Reinforcements

Both zinc oxides used as reinforcements, recycled and commercial ones, were characterized by field emission scanning electron microscopy (FEG-SEM) images. Their morphologies are shown at Figure 2. Commercial ZnO nanoparticles show varied geometries with predominant elongated shapes up to 500 nm. On the other hand, recycled particles have nanoplatelet morphologies with an average thickness of 35 nm. These platelets are grouped, forming desert roses with diameters around 2.5 µm. It is worthy to note that the recycled particles, whose synthesis has been previously published [35], show a homogenous geometry, without any other residual components. This enhances its use as filler.

Figure 3 shows FTIR spectra of both zinc oxides. Compared to commercial zinc oxide, a stronger signal can be observed in recycled around 3430 and 1640 cm^−1^ due to the presence of bonded water. This is associated to the inherent intrinsic moisture of particles obtained by hydrothermal recycling manufacture. FTIR also confirms that recycled ZnO particles have a high purity, without identifying notable rests of other chemicals from recycling of batteries, showing, in addition to the O–H bands, a unique characteristic peak at 420 cm^−1^ associated with the Zn–O bond. In that way, Table 1 notes the chemical composition of recycled ZnO powder determined through XRF in previous research: ZnO contents are above 92 wt %. and only significant quantities of K_2_O and P_2_O_5_ were founded [35]. 

### 3.2. Sedimentation Analysis and Desing for Rotary Moulding

#### 3.2.1. Analysis of Particles Sedimentation

The main source of sedimentation phenomenon is a high difference between the density of polymer matrix and ceramic reinforcements. The sedimentation occurs when mixtures are poured into molds, without forced agitation and external movements for curing, and the viscosity of non-cured resin is considerably low, and its gel times is quite long. Reynolds numbers within polymer matrix are very reduced and flows can be defined by laminar regimes. In these cases, sedimentation of particles can be modelized by the Stokes law [38]:(1)$$v_{s}=\frac{2}{9}\frac{r^{2}g\left(\rho_{p}-\rho_{r}\right)}{\eta }$$
where $v_{s}$ is the sedimentation speed, *r* is the particles equivalent ratio, $\rho_{p}$ and $\rho_{r}$ are the densities of the particles and resin, respectively and η is the viscosity of non-cured resin. Particles agglomeration, after dispersion treatments, is favored in low viscosities mixtures. This behavior increases sharply the reinforcement tendency to sedimentation because the terminal speed maintains an exponential relationship with particle or aggregate sizes. Moreover, the sedimentation speed of each particle or aggregate depends on the gel time of the resin, which is the time to reach a network in gel state. 

In this moment, the sedimentation phenomenon stops. Naturally, the sedimentation is also affected by the particle’s downfall across the thicknesses of the composites:(2)$$d_{s}=v_{s}\cdot t_{gel}$$
where $v_{s}$ is the sedimentation speed, *d_s_* is the thickness of the composite and $t_{gel}$ is the gelation time of resin.

The sedimentation of zinc oxides, or any other particle-shaped reinforcement, can be studied through Equations (1) and (2) and trough the characterization of the particle’s dispersions within the matrices. The polymer matrix used, Araldite LY556, is an aeronautical degree resin with very low viscosity at high temperatures. Its rheological behavior was characterized in previous researches, founding that viscosity descends below 0.1 Pa·s, when temperature increases up to curing set point [39]. XB 3474, as usual in amine hardeners, also presents quite lower viscosities and contributes to further reduce these values. To consider the reinforcement effect, other studies analyzed the increase of the epoxy viscosity with the addition of ZnO particles. In general, it does not grow notably until the weight percentage of reinforcement approaches to 30%. At these concentrations, viscosity increases up to a hundred per cent compared with the neat resin. Beyond this particle concentrations, the viscosity increases exponentially, and the composite begins to be excessively difficult to manufacture by molding [40].

Therefore, the sedimentation phenomenon can be modelized by calculating terminal speed and sedimentation distances as functions of reinforcement equivalent ratios and taking into account the following parameters:(3)$$\rho_{p}=5610\frac{\mathrm{kg}}{m^{3}},\;\rho_{r}=1120\frac{\mathrm{kg}}{m^{3}},\;\eta_{min}=0.05\;\mathrm{Pa}\cdot s,\;\eta_{max}=0.1\;\mathrm{Pa}\cdot s,\;t_{gel}=35\;\min$$

Viscosities values of the mixtures of resin, hardener and particles are settled as medium rates before gelation time. On the other hand, $t_{gel}$ of composites is settled according to experimental measurements. Looking to the referenced bibliography [39], the viscosities values for composites reinforced up to 20 wt % ZnO contents are expected to be closer to the minimum rate and nearest to the top for composites reinforced with contents at 30 wt %. If powder loads rise above this last concentration, it would be expected that the exponential growth of viscosity implies a reduction of sedimentation phenomena to neglectable values. 

At this point, it is worth mentioning that some extended modifications of Equation (1) were purposed by other authors to consider the effect of parameters like volume fraction of powders and dispersion effectiveness within fluids [41,42]. However, it was considered more convenient to define a working region in Figure 4 marked off by maximum and minimum sedimentation curves delimited by lowest and highest viscosities reachable for manufacturable LY556-ZnO dispersions.

The sedimentation speeds and the particles downfalls are shown in Figure 4, confirming that the sedimentation strongly depends on the size of particles. Thus, an experimental analysis of particles dispersion was made. ZnO particles were dispersed in the epoxy monomer by sonication for 1 h. The reinforced matrices were observed to characterize the sonication effectivity and to analyze the size distribution of ceramic particles and agglomerates. Figure 5a,b exhibits the dispersion of non-cured mixtures reinforced with both zinc oxide particles at 6 wt % content. The sonication effectivity for commercial particles into non-cured resin is quite high, so large aggregates are removed, and particles sizes are considered with equivalent ratios smaller than 1 µm, the detection limit of transmission optical microscopy. On the other hand, the dispersions of recycled zinc oxides can be properly characterized by TOM due to its higher particles size. Further, the sonication cycle used is not so efficient with this reinforcement and it can be seen a certain presence of aggregates. Figure 5b shows a histogram with the particles and agglomerates sizes distribution observed. These clear differences of behavior are explained by the different geometry and composition of both studied ceramic particles. 

Although the dimensions of both isolated ceramic fillers, commercial and recycled ones, have nanoscale, the recycled ZnO particles has an important tendency to form micro-scale agglomerations (Figure 2d) due to their inherent moisture (Figure 3). This means that the achieved dispersions are suitable and the best available. The modification of experimental dispersion parameters does not imply better dispersion of recycled particles.

Figure 6 shows SEM images of the cross-section of manufactured composites. Due to the different real size of ceramic particles, the composites reinforced with recycled ZnO suffer more severe sedimentation (Figure 6b,d) compared to composites filled with commercial ZnO (Figure 6a). Figure 6d and 6e show a comparison among theoretical sedimentation (left, calculated by Equations (1) and (2)) and the real particles distribution observed with SEM (right) of epoxy composite reinforced with recycled ZnO. Therefore, it is confirmed that the presented sedimentation model (Figure 4) can rightly determine the generation of a sediment layer that accumulates more than a half of the total reinforcement added to the matrix is expected for recycled ZnO, which is confirmed by SEM. 

Comparing both composites manufactured with different contents of recycled ZnO particles (Figure 6b,c), the thickness of sedimented layer grows in proportion to the added ZnO percentage. However, in both cases, the recycled zinc oxide mass fraction in the sediment layers are near to 70%. This value was determined by image analysis. Therefore, the quantity of particles added into the resin has no noticeable influence on the percentage accumulated in the sediment layer, only affecting its thickness. This saturation of particles in the down sediment layer can be explained by an abrupt local increase of viscosity at the bottom, due to the sedimentation phenomenon, leading to create a resistance against the downfall of the reinforcement. Consequently, a uniform sedimented layer is developed, which contains a constant value near to 70 wt % of ZnO and whose thickness depends on the ZnO content added into the epoxy matrix. 

#### 3.2.2. Design of Rotary Molding 

The experimental results shown confirm that the sedimentation of the studied composites, considering the low viscosity and long gel time of epoxy matrix, strongly depends on the size of particles and agglomerations. In fact, the better dispersion achieved for composites reinforced with commercial ZnO avoids the sedimentation. However, the tendency to agglomerate of recycled ceramic particles, which implies an important increase of their effective filler size, causes an important sedimentation phenomenon. According to Equation (1), the unique mode to avoid the sedimentation phenomena is to remove the gravity effect. In that way, the use of a closed mold that rotates around a concentric axis throughout the curing treatment, at least up to gel time, is analyzed. This concentric rotation prevents the downfall of particles by forcing them to move in conjunction with the mold. 

However, this rotary motion induces centrifugal acceleration to the particles that should be considered to ensure that dispersions remain homogeneous until the resin gelation.
(4)$$v_{c}=\frac{2}{9}\frac{r^{2}a_{c}\left(\rho_{p}-\rho_{r}\right)}{\eta }$$
(5)$$a_{c}=\omega^{2}R$$
where $a_{c}$ is the centrifugal acceleration, ω the angular speed of the mould an *R* the distance between mould and rotation axis.

As shown in Equation (3), the centrifugal speed of particles keeps some similarity with the sedimentation speed Equation (1), changing the action of the gravity by the centrifugal acceleration. Therefore, turning speeds and distances between mold and rotation axis must be reduced to avoid particles movements and maintain the dispersion homogeneity. In the present research, the rotation speed of the mold is fixed at 11 rpm and the distance of the mold with the rotation axis is 10 cm. These experimental parameters generate accelerations near to 0.13 m/s^2^ which induce practically negligible displacements to the ZnO particles during the curing treatment. Figure 7 shows SEM images of a cured composite reinforced with the highest studied content of recycled ZnO manufactured by rotatory molding. The particles distribution is totally homogenous, avoiding the sedimentation phenomenon. 

Compared with composites manufactured by static molding (Figure 6), this alternative method avoids completely the sedimentation and creates homogeneous materials with enhanced matrix-reinforcement interactions. As high magnification (Figure 7b), it is possible to observe the isolated particles, whose size is close to 1–3 µm, similar to initial dispersion achieved by sonication (histogram in Figure 5b). Moreover, a new composite reinforced with 30 wt % of ZnO was manufactured by the designed rotatory molding, confirming homogenous distribution of the ceramic fillers without high agglomerations or sedimentation (Figure 7c). These improved composites could be adequate for insulation applications, with enhanced hydrothermal ageing resistance thanks to the homogeneous presence of ZnO across the matrix. A homogenous ceramic dispersion on the nanocomposites also enhances their wear properties, making them interesting protective coatings with high chemical and wear resistance. 

In summary, to avoid the sedimentation is possible to modify the experimental parameters of materials, such as a reduction of particle size, an increase of resin viscosity or a decrease of gel time. However, for determined materials, the unique mode to avoid it is to modify the molding process, changing the gravity by rotatory forces. In fact, the authors have designed a new method of molding process of composites which allows to fully eliminate the sedimentation phenomena. This method is very suitable to manufacture epoxy resins with recycled ZnO particles from alkaline batteries, giving a new use and application and avoiding the environmental problems of their waste storage.

### 3.3. Chracterization of Composites

#### 3.3.1. FTIR Analysis

The effect of ZnO on epoxy matrix is analyzed through ATR technique. FTIR spectra of epoxy resin and composites do not show significant absorption peaks at 915 cm^−1^ in any sample, indicating that the cure cycles are well designed, and not important amounts of unreacted epoxy rings remain. ZnO addition does not eliminate characteristic peaks of the matrix spectrum but provokes shifts and changes on the intensities. Similar tendency was observed by Mostafaei et al. [43] and Ammar et al. [4] in epoxy-PANI and epoxy-PMDS composites reinforced until 8 wt % ZnO. In the composites made through rotary molding in this research, shifts are clearly observed at 1034–1028 cm^−1^, 1234–1229 cm^−1^ and 3396–3383 cm^−1^ peaks. Additions of ZnO up to 6 wt % displace the peaks to minor wavenumbers, but when the contents rise to 10 and 30 wt %, the trend reverses and peaks wavenumber become closer to the net resin values.

Attenuation on –OH bands caused by ZnO are significant. Ghule et al. [44] and Ammar et al. [4] reported this phenomenon and argued that sonication favors the interaction between hydroxyl functional groups and ZnO particles. In that way, the bands tend to disappear with the increase of the ZnO content. This effect is deeper in recycled ZnO composites, which indicates that sonication cycles are well realized, and rotary molding keeps a fine distribution that helps to enhance the interaction between matrix and reinforcement. At 30 wt % content, the manufacturing of recycled ZnO composite continues achieving this distribution without the formation of aggregates, as seen in Figure 8, and the –OH band disappear almost completely. However, in the case of 30 wt % commercial ZnO composite, the spectrum shows a notable increase on this peak. This opposite behavior can be explained by the excessive viscosity of the mixture, due to the greater particles surface area and high load, which may hinder the sonication effectivity.

#### 3.3.2. Thermo-Mechanical Behavior 

Once the manufacture of epoxy composites doped with recycled and commercial ZnO particles has been optimized using the rotary molding, their mechanical and thermal properties has been analyzed. DMTA analysis was used to study the thermo-mechanical properties of composites. Figure 9 shows the average values for the storage modulus at room temperature, which are associated to the stiffness, and glass transition temperature (*T*_g_), whose value indicates the maximum operating temperature, of manufactured composites as a function of ZnO content.

As expected, the addition of ZnO particles achieve increases on glass transition temperatures of composites due to the steric hindrance of ceramic particles on chain segments movements and its capability to catalyze the curing reaction through acid-base interactions with hardener amine groups. Moreover, the inner moisture present in recycled zinc oxide structure seems to enhance the interaction with the polymer matrix compared with commercial ZnO and brings on its thermal stability with higher values of *T*_g_ [22,23]. This effect takes special relevance in medium filler loads. As the addition of reinforcement rises to greater values, the full occupation of the free volume of thermosetting matrix limits the enhancement and then the glass transition temperature of ZnO-epoxy composites tends to decrease [45]. Similar behavior at lower particle loads for epoxy/ZnO nanocomposites was observed by Zabihi et al [22]. Recently, Ponnamma et al. [13] revised the influence of ZnO particles in the glass transition of different polymers. They confirmed that *T*_g_ of matrix and crosslinking could decrease due to imperfect dispersions or aggregate formations, even at contents under 2 wt % [4]. 

In addition to SEM images shown in Figure 7 and FTIR analysis, the thermomechanical properties obtained by DMTA analysis confirm the high-quality reinforcement dispersions achieved by curing with rotational molding: the addition of ZnO still improving the epoxy properties with loads closer to 10 wt %. 

The storage modulus notably increases with ZnO percentage for both studied particles. It is worthy to note that, despite of the lower crosslinking degree of their matrix the composites reinforced with commercial ZnO nanoparticles, they achieve similar stiffness than the epoxy resins doped with recycled ones, especially at medium loads. This effect could be explained by the real nano-scaled size of commercial nanoparticles, which implies a higher surface area, inducing higher interaction areas with the polymer matrix.

#### 3.3.3. Microhardness

One of the objectives to add ceramic particles into thermosetting resins is the enhancement of their hardness. Microhardness of the manufactured composites are noted in Table 2. Vickers microhardness of the used neat aeronautical resin is close to 22. The addition of ZnO particles induces a light increment of hardness without, however, being clearly influenced by the different geometry of ceramic particles [15]. The reason for this is that the composite surface remains unchanged, as is confirmed by SEM images of cross-section. There is an outer micro-scale layer of epoxy resin, which is tested during the hardness test. Nevertheless, at the highest recycled ZnO content (30 wt %), the hardness is enhanced by up to 80%. This result implies that ceramic particles appear massively on the surface of the epoxy composites when high contents are added.

## 4. Conclusions

In this work, a new use of zinc oxide particles from spent alkaline batteries is developed, avoiding environmental and health problems. Thousands of tons of alkaline batteries are recycled across the world each year. 

Recycled ZnO particles present an inner moisture content, which enhances the interaction with epoxy matrix, obtaining composites with enhanced thermal and mechanical properties. 

The recycled particles have a specific geometry, shaped as desert roses with an average diameter of 2.5 µm. These dense ceramic particles suffer sedimentation during the curing treatment of epoxy resin. The experimental parameters that influence the sedimentation phenomenon are studied, such as the low viscosity of non-cured resin, the relative high gel time to form a crosslinked network and the size of added particles. If any material can be modified, in principle, the composites with recycled ceramic particles could not be manufactured by conventional processes. For it, a new manufacturing method has been successfully developed using a rotatory mold to avoid the sedimentation. This allows for the manufacturing of homogenous composites which achieve thermomechanical properties enhancements with higher ceramic filler contents than conventional methods. In summary, the composite reinforced with 30 wt % recycled ZnO particles presents enhancements in its glass transition temperature (1.4%), its stiffness (19.2%) and hardness (82.3%). Moreover, this method results in an easily applicable way to optimize any other reinforced polymer networks with positive results, as demonstrated with the epoxy–commercial ZnO composites.

## Figures and Tables

**Figure 1 polymers-12-01619-f001:**
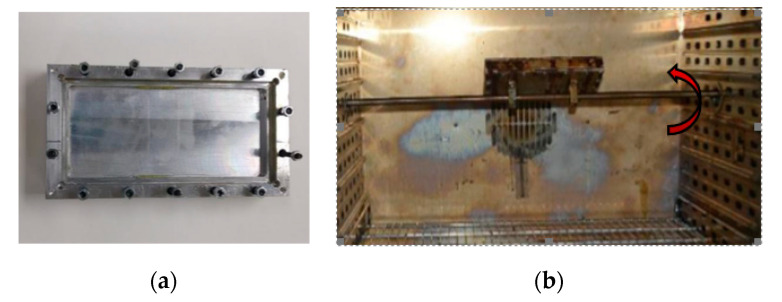
Rotary mold used for composites manufacturing. (**a**) View of the disassembled rotary mold; (**b**) Rotary mold working inside the oven.

**Figure 2 polymers-12-01619-f002:**
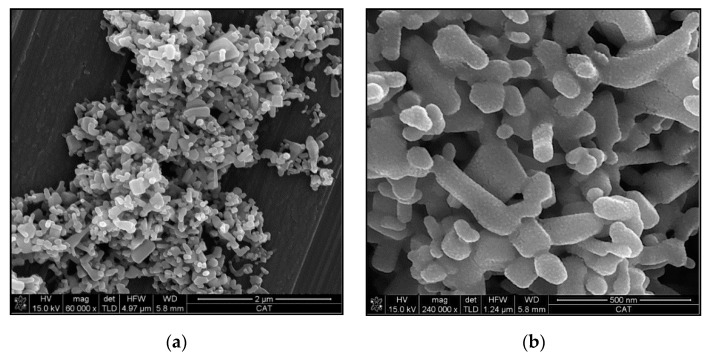
FEG-SEM images of commercial (**a**,**b**) and recycled zinc oxide (**c**,**d**). Write square (**a**,**c**) indicates the magnified area (**b**,**c**).

**Figure 3 polymers-12-01619-f003:**
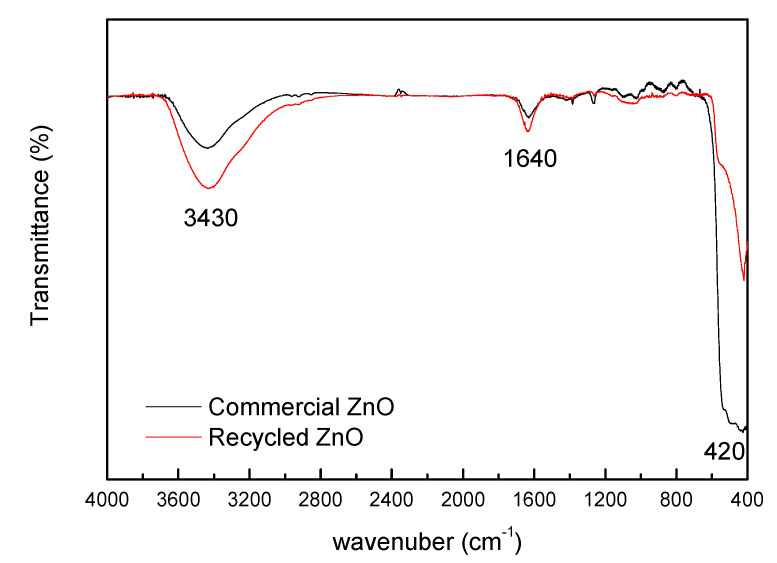
Commercial and recycled zinc oxides FITR spectra.

**Figure 4 polymers-12-01619-f004:**
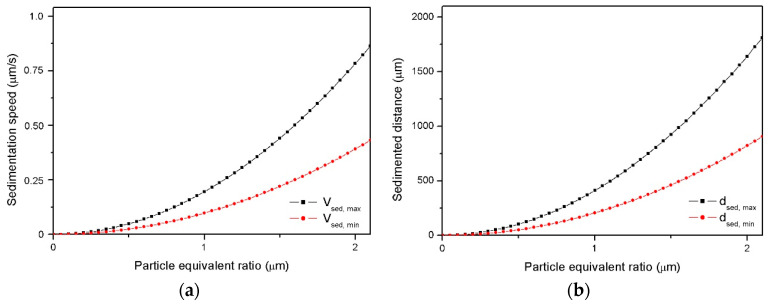
Theoretical sedimentation speed of ZnO within LY 556 (**a**) and sedimented distance before gelation (**b**).

**Figure 5 polymers-12-01619-f005:**
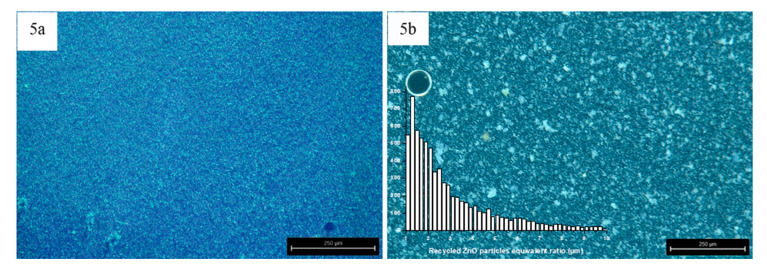
TOM images of the dispersion of non-cured mixtures with 6 wt % commercial ZnO (**a**) and recycled ZnO and histogram with its particles and aggregates size distribution (**b**).

**Figure 6 polymers-12-01619-f006:**
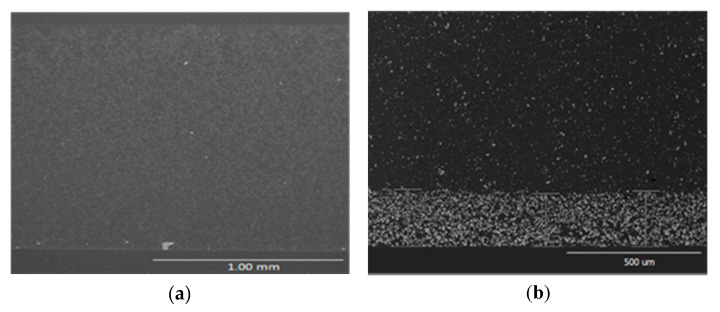
SEM images of cross-section of cured composites reinforced with 6 wt % of commercial ZnO (**a**) and 6 wt % (**b**) and 10 wt % (**c**) of recycled ZnO. A comparison among theoretical sedimentation (left) and the real particles distribution observed with SEM at high magnification (right) of epoxy composite reinforced 6 wt % (**d**) and 10 wt % (**e**) of recycled ZnO.

**Figure 7 polymers-12-01619-f007:**
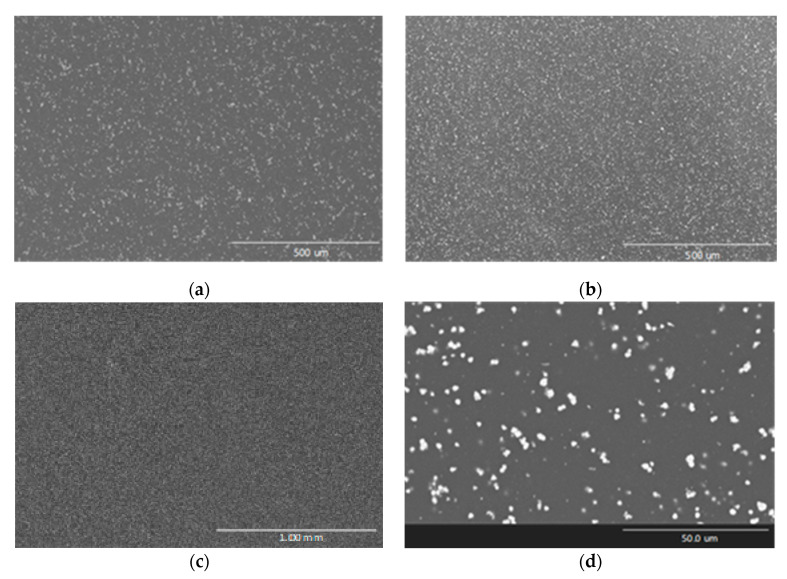
SEM images of cross-section of cured composites reinforced with 6 (**a**), 10 (**b**) and 30 wt % (**c**) recycled ZnO manufactured by rotary molding. Amplified details are observed at high magnification (**d**).

**Figure 8 polymers-12-01619-f008:**
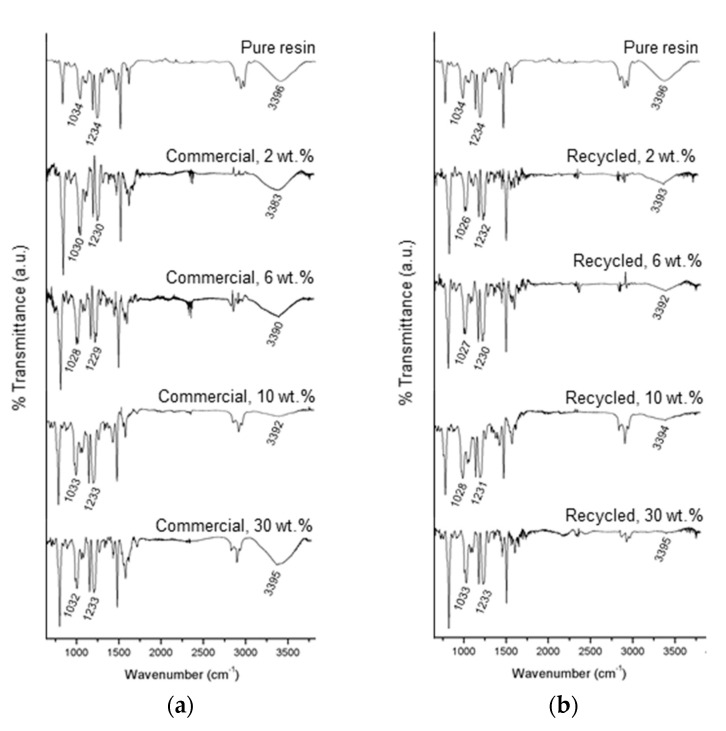
FTIR spectra of epoxy resin, commercial ZnO reinforced composites (**a**) and recycled ZnO reinforced composites (**b**).

**Figure 9 polymers-12-01619-f009:**
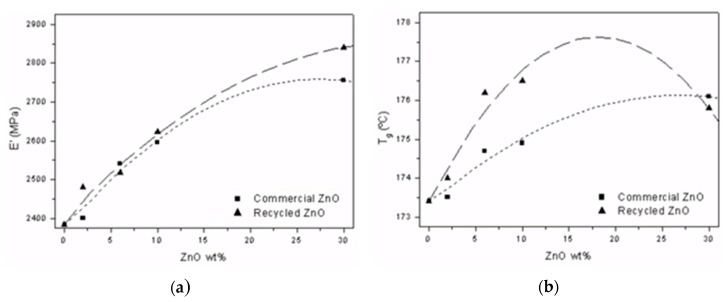
Storage modulus at 30 °C (**a**) and glass transition temperature (**b**) of ZnO/epoxy composites.

**Table 1 polymers-12-01619-t001:** Recycled ZnO chemical composition.

MgO	Al_2_O_3_	SiO_2_	P_2_O_5_	Cl	K_2_O	CaO	TiO_2_	MnO	Fe_2_O_3_	NiO	CuO	ZnO
0.4	0.09	0.16	1.48	0.03	4.20	0.34	0.03	0.02	0.10	0.11	0.17	92.73

**Table 2 polymers-12-01619-t002:** Microhardness values of neat epoxy and epoxy-ZnO composites.

	0 wt %	2 wt %	6 wt %	10 wt %	30 wt %
Recycled ZnO	22.0 ± 1.4	22.6 ± 3.0	22.4 ± 1.0	23.2 ± 1.9	40.1 ± 6.9
Commercial ZnO	22.0 ± 1.4	22.0 ± 0.8	22.0 ± 0.7	24.3 ± 2.8	37.3 ± 8.7

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
