# Peer review of "New Manufacturing Process of Composites Reinforced with ZnO Nanoparticles Recycled from Alkaline Batteries"

_polymers, 2020, doi:10.3390/polym12071619_

Round 1

Reviewer 1 Report

Dear Editor, and Dear Authors

The paper titled  “New Manufacturing Process of Composites Reinforced with ZnO Nanoparticles Recycled from Alkakine Batteries” addresses the investigation of particles sedimentation during thermosetting processing through rotary moulding, and the reinforcement effects of zinc oxide obtained from alkaline batteries recycling. The paper objective is quite interesting and the whole manuscript is well written and presented, therefore the paper may be accepted for publication after minor revision. Please see the below comments.

Additional comments:

  • Methodology: please inform the recycling procedure applied to produce ZnO from alkaline batteries. Please inform FTIR applied procedure.
  • Results and Discussion: Have the authors determined ZnO purity (besides FTIR)? Which are the main intended applications for the produced epoxy compounds? Please improve the quality of Figure 4.

Best regards,

Author Response

The paper titled “New Manufacturing Process of Composites Reinforced with ZnO Nanoparticles Recycled from Alkakine Batteries” addresses the investigation of particles sedimentation during thermosetting processing through rotary moulding, and the reinforcement effects of zinc oxide obtained from alkaline batteries recycling. The paper objective is quite interesting and the whole manuscript is well written and presented, therefore the paper may be accepted for publication after minor revision.

Thank you for your comments about the article. Below you can find detailed answer to each of the specific comments.

Please see the below comments

Methodology: Please inform the recycling procedure applied to produce ZnO from alkaline batteries.

We completely agree about this aspect. The procedure of the synthesis of ZnO nanoparticles from recycling of alkaline batteries has been extended.

 In the revised manuscript, the additional information has been added in red.

The synthesis of this second type of zinc oxide consists on a three stages process:

  1. Leaching of batteries black mass: black mass was leached by adding 1.72 mol/L of (NH4)2CO3 and 0.5 mol/L of NH3. Black mass pulp concentration was settled at 100 g/L and the solution was agitated at 1800 rpm under ambient conditions for 1 hour. The after-treatment suspension obtained was filtered under vacuum and leachate was kept for further processing.
  2. Leachate evaporation under vacuum (34 mbar) and drying of the solid precursor gotten in an oven at 80⁰C for 6 hours.
  3. Calcination of the post-evaporation solid precursor at 800⁰C during 5 h to obtain the ZnO.

Methodology: Please inform FTIR applied procedure

 In the revised manuscript, the additional information has been added in red.

 ZnO particles were analyzed by Fourier transform infrared spectroscopy (FTIR), using a Varian 670 FTIR spectrometer (spectral range 4000-400 cm−1, spectral resolution of 4 cm−1) in transmittance mode. This analysis was performed using KBr pellets with 1 wt.% of ZnO. Epoxy-ZnO composites were also analyzed by attenuated total reflection technique (ATR, spectral range 3800-700 cm−1).

 Results and Discussion: Have the authors determined ZnO purity (besides FTIR)?

The authors agree with the reviewer. The information about the purity of the recycled ZnO nanoparticles is added in the new revised manuscript. The chemical composition was measured by XRF. The purity of the commercial nanoparticles is supplied by the manufacturer and included in the methodology part.

In that way, Table 1 notes the chemical composition of recycled ZnO powder determined through XRF in previous research: ZnO content are above 92 wt.%. and only significant quantities of K2O and P2O5 were founded [36].  

Table 1. Recycled ZnO chemical composition.

MgO

Al2O3

SiO2

P2O5

Cl

K2O

CaO

TiO2

MnO

Fe2O3

NiO

CuO

ZnO

0.4

0.09

0.16

1.48

0.03

4.20

0.34

0.03

0.02

0.10

0.11

0.17

92.73

 Results and Discussion: Which are the main intended applications for the produced epoxy compounds?

 In the revised manuscript, it is included the information about the intended applications of manufactured composites as insulating materials with enhanced chemical resistance.

These improved composites could be adequate for insulation applications, with enhanced hydrothermal ageing resistance thanks to the homogeneous presence of ZnO across the matrix. A homogenous ceramic dispersion on the nanocomposites also enhances their wear properties, making them interesting protective coatings with high chemical and wear resistance.  

 Results and Discussion: Please improve the quality of Figure 4.

Thank you very much. The quality if Figure 4 has been improved in the revised manuscript.

Reviewer 2 Report

This paper described the manufacturing method of thermosetting resins reinforced with dense particles developed. As a result, the author reported the used reinforcement zinc oxide obtained by new recycling method from alkaline batteries and the recycled ZnO particles enhanced the interaction of epoxy matrix due to their inner moisture. I recommend this paper to be published after minor revision since the meaningful results were reported in this field.

(1) The dispersion information of ZnO particles were added in detail in the revised text.

(2) The interaction between epoxy resin and ZnO was analyzed in detail via FT-IR in the revised text.

(3) In Introduction part, the related refs for ZnO were additionally inserted.

Author Response

This paper described the manufacturing method of thermosetting resins reinforced with dense particles developed. As a result, the author reported the used reinforcement zinc oxide obtained by new recycling method from alkaline batteries and the recycled ZnO particles enhanced the interaction of epoxy matrix due to their inner moisture. I recommend this paper to be published after minor revision since the meaningful results were reported in this field.

Thank you very much for your comment.

The dispersion information of ZnO particles were added in detail in the revised text.

This new information is added in the revised manuscript in red.

To create the composites, the epoxy resin was heated up to 50⁰C to reduce its viscosity and then the ZnO particles were added and mixed mechanically. To achieve a homogeneous distribution, the particles were dispersed afterwards into the epoxy matrix by sonication with a cycle of 50 Hz and 50% of amplitude during 1 h. This was done by using a 7 mm diameter probe sonicator. To control the temperature during sonication and avoid overheating capable of triggering epoxy homopolymerization, cold water baths were placed around the epoxy-ZnO mixtures. Afterwards, the dispersion was degassed 15 min at 80ºC. Then, the hardener was added in a stoichiometric ratio and finally the non-cured mixture was placed in a mold for curing at 140ºC for 8 hours.

The interaction between epoxy resin and ZnO was analyzed in detail via FT-IR in the revised text.

This comment is very interesting. In the revised manuscript, the authors have included an extended analysis of the FTIR spectra of nanocomposites in order to evaluate the chemical interaction between epoxy matrix and ZnO nanoparticles. The quality of the paper has been enhanced with this incorporation.

3.3.1. FTIR analysis

 Effect of ZnO on epoxy matrix is analyzed through ATR technique. FTIR spectra of epoxy resin and composites do not show significant absorption peaks at 915 cm-1 in any sample, indicating that the cure cycles are well designed, and not important amounts of unreacted epoxy rings remain. ZnO addition does not eliminate characteristic peaks of the matrix spectrum but provokes shifts and changes on the intensities. Similar tendency was observed by Mostafaei et al. [44] and Ammar et al. [4] in epoxy-PANI and epoxy-PMDS composites reinforced until 8 wt.% ZnO. In the composites made through rotary molding in this research, shifts are clearly observed at 1034-1028 cm-1, 1234-1229 cm-1 and 3396-3383 cm-1 peaks. Additions of ZnO up to 6 wt.% displace the peaks to minor wavenumbers, but when the contents rise to 10 and 30 wt.%, the trend reverses and peaks wavenumber become closer to the net resin values.

Attenuation on -OH bands caused by ZnO are significant. Ghule et al. [45] and Ammar et al. [4] reported this phenomenon and argued that sonication favors the interaction between hydroxyl functional groups and ZnO particles. In that way, the bands tend to disappear with the increase of the ZnO content. This effect is deeper in recycled ZnO composites, which indicates that sonication cycles are well realized, and rotary molding keeps a fine distribution that helps to enhance the interaction between matrix and reinforcement. At 30 wt.% content, manufacturing of recycled ZnO composite still achieving this distribution without aggregates formation, as seen in Fig. 7, and the -OH band disappear almost completely. However, in case of 30 wt.% commercial ZnO composite, the spectrum shows a notable increase on this peak. This opposite behavior can be explained by the excessive viscosity of the mixture, due to the greater particles surface area and high load, which may hinder the sonication effectivity.

Figure 8. FTIR spectra of epoxy resin, commercial ZnO reinforced composites (8a) and recycled ZnO reinforced composites (8b).

 (3) In Introduction part, the related refs for ZnO were additionally inserted

[10]       H. Zhou et al., “Fabrication of ZnO/epoxy resin superhydrophobic coating on AZ31 magnesium alloy,” Chem. Eng. J., vol. 368, no. January, pp. 261–272, 2019.

 [11]      N. Wang, L. Tang, Y. Cai, W. Tong, and D. Xiong, “Scalable superhydrophobic coating with controllable wettability and investigations of its drag reduction,” Colloids Surfaces A Physicochem. Eng. Asp., vol. 555, no. July, pp. 290–295, 2018.

 [18]      P. K. Sandhya, M. S. Sreekala, M. Padmanabhan, and S. Thomas, “Mechanical and thermal properties of ZnO anchored GO reinforced phenol formaldehyde resin,” Diam. Relat. Mater., vol. 108, no. May, p. 107961, 2020.

 [19]      M. S. Ghamsari, S. Alamdari, D. Razzaghi, and M. Arshadi Pirlar, “ZnO nanocrystals with narrow-band blue emission,” J. Lumin., vol. 205, no. September 2018, pp. 508–518, 2019.

 [20]      S. A. Hawkins, H. Yao, H. Wang, and H. J. Sue, “Tensile properties and electrical conductivity of epoxy composite thin films containing zinc oxide quantum dots and multi-walled carbon nanotubes,” Carbon N. Y., vol. 115, pp. 18–27, 2017.

 [21]      A. M. Kumar, A. Khan, M. Y. Khan, R. K. Suleiman, J. Jose, and H. Dafalla, “Hierarchical graphitic carbon nitride-ZnO nanocomposite: Viable reinforcement for the improved corrosion resistant behavior of organic coatings,” Mater. Chem. Phys., vol. 251, no. October 2019, p. 122987, 2020.

 [24]      M. Ghaffari, M. Ehsani, M. Vandalvand, E. Avazverdi, A. Askari, and A. Goudarzi, “Studying the effect of micro- and nano-sized ZnO particles on the curing kinetic of epoxy/polyaminoamide system,” Prog. Org. Coatings, vol. 89, pp. 277–283, 2015.

 [25]      K. K. Jena, S. M. Alhassan, and N. Arora, “Facile and rapid synthesis of efficient epoxy-novolac acrylate / MWCNTs-APTES-ZnO hybrid coating films by UV irradiation: Thermo-mechanical, shape stability, swelling, hydrophobicity and antibacterial properties,” Polymer (Guildf)., vol. 179, no. June, p. 121621, 2019.

 [26]      J. Ma, W. An, Q. Xu, Q. Fan, and Y. Wang, “Antibacterial casein-based ZnO nanocomposite coatings with improved water resistance crafted via double in situ route,” Prog. Org. Coatings, vol. 134, no. June 2018, pp. 40–47, 2019.

 [30]       F. Y. Yuan, H. Bin Zhang, X. Li, X. Z. Li, and Z. Z. Yu, “Synergistic effect of boron nitride flakes and tetrapod-shaped ZnO whiskers on the thermal conductivity of electrically insulating phenol formaldehyde composites,” Compos. Part A Appl. Sci. Manuf., vol. 53, pp. 137–144, 2013.

 [31]      Y. Jiang, R. Sun, H. Bin Zhang, P. Min, D. Yang, and Z. Z. Yu, “Graphene-coated ZnO tetrapod whiskers for thermally and electrically conductive epoxy composites,” Compos. Part A Appl. Sci. Manuf., vol. 94, pp. 104–112, 2017.

 [33]      H. Yari and M. Rostami, “Enhanced weathering performance of epoxy/ZnO nanocomposite coatings via functionalization of ZnO UV blockers with amino and glycidoxy silane coupling agents,” Prog. Org. Coatings, vol. 147, no. February, p. 105773, 2020.
